# Do NSm Virulence Factors in the *Bunyavirales* Viral Order Originate from Gn Gene Duplication?

**DOI:** 10.3390/v16010090

**Published:** 2024-01-05

**Authors:** Victor Lefebvre, Ravy Leon Foun Lin, Laura Cole, François-Loïc Cosset, Marie-Laure Fogeron, Anja Böckmann

**Affiliations:** 1MMSB—Molecular Microbiology and Structural Biochemistry, Université de Lyon 1, CNRS UMR 5086, F-69367 Lyon, France; victor.lefebvre@etu.univ-lyon1.fr (V.L.); ravy2.leonfounlin@gmail.com (R.L.F.L.);; 2CIRI—Centre International de Recherche en Infectiologie, Université de Lyon 1, ENS de Lyon, Inserm U1111, CNRS UMR 5308, F-69007 Lyon, France

**Keywords:** *Bunyavirales*, virulence factor, NSm, AlphaFold, structure

## Abstract

One-third of the nine WHO shortlisted pathogens prioritized for research and development in public health emergencies belong to the *Bunyavirales* order. Several *Bunyavirales* species carry an NSm protein that acts as a virulence factor. We predicted the structures of these NSm proteins and unexpectedly found that in two families, their cytosolic domain was inferred to have a similar fold to that of the cytosolic domain of the viral envelope-forming glycoprotein N (Gn^cyto^) encoded on the same genome fragment. We show that although the sequence identity between the NSm^cyto^ and the Gn^cyto^ domains is low, the conservation of the two zinc finger-forming CysCysHisCys motifs explains the predicted structural conservation. Importantly, our predictions provide a first glimpse into the long-unknown structure of NSm. Also, these predictions suggest that NSm is the result of a gene duplication event in the *Bunyavirales Nairoviridae* and *Peribunyaviridae* families and that such events may be common in the recent evolutionary history of RNA viruses.

## 1. Introduction

Virology research is often conducted in a situation of urgency, going from knowing virtually nothing about a virus to understanding it as quickly as possible. To change this inefficient situation, the World Health Organization (WHO) has decided on a list of ten pathogens for which research should be prioritized. These pathogens have a high pandemic potential due to their lethality and the lack of effective treatment or prevention. They are mainly transmitted by vectors such as mosquitoes and ticks, with which humans will increasingly interact in the current context of climate change. *Bunyavirales* thus represent a persistent global health challenge, and research on these pathogens is important to prevent future outbreaks or even an increase in their spread, which could have devastating consequences.

The *Bunyavirales*, a large viral order, provide three of these pathogens (the Lassa (LAS), Rift Valley Fever (RVF), and Crimean–Congo Hemorrhagic Fever (CCHF) viruses). Although research on these viruses is now intensifying at all levels, it still lags far behind that on other shortlisted viruses. A virulence factor present in many pathogens of the order, the non-structural protein m (NSm), has roles in antagonizing immune responses, promoting viral assembly and infectivity, and maintaining infection in their transmission vectors [1]. Although a pathogenic virus in the order may not have such an NSm protein and a virus containing an NSm is not necessarily a pathogen, NSm has been shown to play a clear role as an amplifier of infection in some of the most threatening viruses, such as CCHFV. Such proteins are thus prime drug targets, and attenuated viruses carrying deletions of these proteins can serve as a basis for vaccine development. Knowledge of the structure of the NSm protein is therefore central to understanding disease progression and viral pathogenesis and to developing strategies for intervention and treatment.

The viruses of the order recognize a wide variety of hosts, including humans, in which they can cause severe diseases. Arthropod vectors such as ticks, mosquitoes, and sand flies provide the most common modes of transmission, which can also be direct, by contact with infectious blood or body fluids (Figure 1a). The *Bunyavirales* are diverse but share a single-stranded negative-polarity RNA genome, often with the three segments L, M, and S (Figure 1b). They encode mostly four structural proteins that make up the viral particle: the two envelope glycoproteins, the NP nucleoprotein, and the L viral, RNA-dependent RNA polymerase. The structural and functional similarities in bunyaviruses have been reviewed recently, in order to evaluate perspectives for pan-bunya antivirals [2].

NSm is, ex vivo, an accessory protein [1] encoded on the M segment in the *Nairoviridae*, *Peribunyaviridae*, *Phenuiviridae* and *Topsoviridae* families of the order [3]. For CCHFV (*Nairoviridae* family) and Bunyamvwera virus (BUNV, *Peribunyaviridae* family), NSm is the C-terminal cleavage product of Pre-Gn [4,5] (Figure 1c). In the *Phenuiviridae* family, as in RVFV [1], and also the *Tospoviridae*, NSm precedes Gn. NSm has 150–200 residues, has been described to be twice membrane-spanning, and localizes mainly to the Golgi in infected cells [6]. The structure of NSm has long remained a mystery. Today, as we show here, artificial intelligence structure prediction programs such as AlphaFold2 [7] can provide a first insight.

## 2. Materials and Methods

An implementation of AlphaFold2.1 [7] software on a local server was used for predictions. Amino acid sequences were entered, and predictions were run in the monomer mode. Amino acid sequences were obtained from the UniProt data base. The resulting AlphaFold archives are provided upon request. Per-residue estimates of the confidence of the models are presented on a scale from 0 to 100 in the individual PDB files of the predictions (pLDDT values). Color coding in the Figures changes from darkblue (100), to light blue (90), yellow (70), orange (50), and red (0). pLDDT > 90 indicates a model with high accuracy, values of 70 and 90 suggest a generally good backbone prediction, values of 50 and 70 indicate low confidence, and values <50 are not reliable. The pLDDT values for the different models are summarized in Appendix A. The predicted structures were visualized and also matched where applicable, using ChimeraX-1.5 [8], which was also used to calculate the backbone root-mean-square deviation (RMSD). Sequence alignments were performed using the multiple protein sequence alignment tools available through the NPS@ web interface [9].

## 3. Results

### 3.1. Full Conservation of the Gn^cyto^ Double Zinc Finger Was Predicted for the Nairoviridae NSm

First, we used AlphaFold to predict the structures of NSm for the *Nairoviridae* family. Figure 2a shows the AlphaFold2 prediction of CCHFV NSm. The resulting model showed two transmembrane domains and one globular domain. As can be seen from color coding, the cytosolic domain showed high accuracy, while the prediction of the transmembrane domains showed lower confidence. The cytosolic domain surprisingly showed a very well-defined fold, predicted with high confidence. A closer look at the AlphaFold2 hit report interestingly showed that the previously determined CCHFV Gn cytosolic domain (Gn^cyto^) structure (PDB 2L7X [10]) represented the best hit in hhsearch (as provided in the AlphaFold2 result files). A comparison of CCHFV NSm and Gn^cyto^ (in magenta in Figure 2a) showed that Gn^cyto^ appeared to be indeed fully reproduced in the predicted NSm model (backbone RMSD of 1.6 Å). An alignment of the CCHFV Gn and NSm cytosolic domain sequences from different strains (Figure 2b and Appendix A) revealed the rationale behind the high similarity: while less than 22% of the sequence is identical, the CysCysHisCys motifs of the two ββα zinc fingers (ZFs) in Gn^cyto^ are indeed fully conserved between the two proteins, as also shown in the zoomed image in Figure 2c. One should also note the fact that neither insertions nor deletions were observed. The few other conserved amino acids are shown in grey; the basic LysArgLys motif, which might play a role in RNA binding, is shown in blue. We compared CCHFV NSm with NSm of another member of the *Nairoviridae* family. Figure 2d shows the alignment (see also Appendix A) and AlphaFold2 models of CCHFV and Dugbe virus (DUGV) NSm^cyto^, which overlap nearly perfectly (RMSD of 1.8 Å).

### 3.2. NSm and Gn^cyto^ of the Peribunyaviridae Family Align Partially

Next, we aligned the sequences and structures of the cytosolic domains of NSm and Gn from the *Peribunyaviridae* family. The *Peribunyaviridae* family also carries NSm in its M gene (Figure 1c). The sequence alignment (Figure 2e; for the full alignment, see Appendix A) revealed the full conservation of seven cysteines and two histidines in Gn^cyto^ and of six cysteines and two histidines in NSm^cyto^. Sequence identity between Gn and NSm is with 19% similar to the one observed in the *Nairoviridae*. In contrast to the *Nairoviridae* Gn^cyto^ structure, the experimental structure of *Peribunyaviridae* Gn^cyto^ is unknown. The pattern that can be postulated in Gn^cyto^ as a C-terminal zinc finger (ZF2, right grey box) is mainly conserved in NSm, resulting in a possible CysCysHisCys motif. The situation is less obvious for the pattern corresponding to a putative N-terminal zinc finger (ZF1), which in Gn^cyto^ could be CysCysCysHis, CysCysHisCys, or even CysCysCysCys, while it must be CysCysHisCys in NSm. The predicted AlphaFold2 models for BUNV Gn and NSm are shown in Figure 2f,g. A double zinc finger was predicted for both Gn^cyto^ and NSm, with two CysCysHisCys motifs. ZF2 in Gn^cyto^ shows a ββα fold like CCHFV Gn^cyto^, while ZF1 is similar to that in HIV-1 NCp7 [11], which also shows a CysxxCysxxxxHisxxxxCys sequence. The RMSD between the two models was 5.8 Å. The resulting models of the other aligned family members appeared very similar (not shown), with the exception of that for AKAV NSm, which presented a very different fold, probably due to the six-residue deletion in ZF2, and was predicted with low confidence (Figure 2h). Thus, in contrast to what was observed for *Nairoviridae*, the predicted structures for Gn^cyto^ and NSm^cyto^ seemed to differ at least partially from each other. The Gn^cyto^ ZF2 ββα fold was surprisingly not found in NSm, despite the very similar length of the loop between the CysCys and the HisCys parts of the motif. ZF1 resulted to be completely different, as already suggested by the different sequence motifs. However, it must be remembered that, in contrast to *Nairoviridae*, for which the Gn^cyto^ structure is known [10], there is no experimental structure for either protein that could assist AlphaFold2 predictions for this family. Thus, to obtain experimental data will be decisive in further analyses.

### 3.3. The NSm Proteins of the Phenuiviridae and Tospoviridae Families Were Predicted to Differ from Gn^cyto^

The third family carrying an NSm protein is the *Phenuiviridae* family, a family of viruses that infect animals, plants, and fungi. The NSm sequences do not suggest the presence of a zinc finger, as only two conserved cysteine residues, and no histidine, are present in RVFV, Toscana virus (TOSV), Arumowot virus (AMTV), and Punta Toro virus (PTPV) NSm proteins (Appendix A). The identity in the alignment between the different NSm proteins appeared to be less than 3%. AlphaFold2 predictions, however, surprisingly led to similar structural models of the NSm proteins, as shown in Figure 2i,j, where the TOSV NSm model is shown in isolation and then superimposed on the three other structural models. In addition to the TOSV NSm structure, the structure of PTPV NSm was predicted quite reliably, whereas those of RVFV and AMTV NSm were judged to be less accurate. The resulting folds include two α-helices and a six-stranded β-sheet. The pairwise RMSD values for the proteins were 2.3 Å (PTPV to TOSV), 4.4 Å (RVFV to TOSV), and 10.0 Å (AMTV to TOSV). Accordingly, Figure 2k shows a manual alignment of the protein sequences taking into account the secondary structural features of the predicted models. The two cysteines remain approximately aligned but do not form a disulfide bond with each other. While the folds look similar, the fact that there is no experimental structure for the different NSm proteins of the *Phenuiviridae* also seemed to result in the lack of a clear model to use in the predictions. Experimental data will be crucial here as well.

Next, we investigated a possible link between NSm and Gn^cyto^. Gn^cyto^ in *Phenuiviridae* was predicted to comprise multiple α-helices, forming a not very well-defined 3D fold (Figure 2l). Identity between the different Gn^cyto^ domains was found to be less than 5% (Appendix A). While the orientation of the helices with respect to each other is not maintained in the different proteins, the α-helical structures are rather consistent. Still, one can see that the reliability, in general, was poor, as indicated by the color coding, besides the light blue helical fragment shown in the center, on which the structures were matched. The predicted models clearly indicated that there is no relationship between NSm and Gn^cyto^, as also reflected in the alignment, which showed less than 10% identity between them (Appendix A).

In *Tospoviridae* that infect plants, such as Tomato Spotted Wilt Virus (TSWV), NSm is called the movement protein and is an important virulence factor [12]. NSm is encoded in the positive strand of the M RNA, using a different strategy than in other *Bunyaviridae*, which may be an adaptation to the plant host. Figure 3a shows for TSWV that NSm was predicted to form a β-sheet motif between residues 100 and 240 and that at both the N- and the C-termini, mainly helical structures were predicted, which showed different orientations with respect to the β-sheet motif. While the prediction accuracy for the β-barrel was high, the N- and C-terminal models are less reliable. The β-sheet motif was found to include the previously identified [13] conserved Pro/Asp-Leu-X, Asp-, and the homologous to phospholipase A2 (PLA2) motifs. From the model for the Gn cytosolic domain in Figure 3b, it can be seen that this domain was predicted to form a double CysCysHisCys zinc finger motif (Appendix A) as in *Nairo*- and *Peribunyaviridae*, but also with modest accuracy. It is clear from our analysis that NSm and the cytosolic domain of Gn showed no similarity in this family (for an alignment between G1/2 and NSm, see Appendix A).

## 4. Discussion

The structure predictions of the different cytosolic domains of Gn and NSm revealed a common fold in two out of the four families investigated: the *Nairoviridae* and the *Peribunyaviridae*. This observation correlates with the position of NSm in the genome, which follows Gn in these two families, but precedes Gn in the third one, the *Phenuiviriade* family (Figure 1c), and in the fourth one, the *Tospoviridae* family. It is unclear how NSm originated; the predicted structural similarity, which is most striking in *Nairoviridae* and partial in *Peribunyaviridae*, leads us to hypothesize that NSm may have originated there from a duplication event of a gene fragment from Gn^cyto^. Gene duplication is a process by which a genetic sequence, or a fragment thereof, is copied, creating an additional gene sequence [14]. This can occur naturally through mutations in DNA replication or through recombination. Both mechanisms that can lead to the formation of gene duplicates, as well as the fates of the new duplicated genes, are wide-ranging and can depend on several factors [15]. The duplication of genes or gene fragments can lead to the evolution of new functions of the duplicated fragment or to an increase in the amount of protein produced by it [14]; both outcomes can result in increased virulence. In viruses, the frequency of gene duplication can vary greatly depending on the specific virus and the conditions under which it replicates [16]. However, the duplication of gene fragments is common in many viruses. In particular, RNA viruses can undergo frequent duplication of gene fragments due to their rapid replication cycles, recombinational potential, and error-prone replication mechanisms. And indeed, previously, a duplication event of a gene fragment from Gn was proposed to have resulted, in CCHFV, in the virulence factor GP38 [17], and an internal gene fragment duplication was described within the CCHFV Gc [18]; for both GP38 and Gc, experimental 3D structures were described. Gene duplication in viruses remains indeed difficult to be established based solely on sequence information. We here showed that structural knowledge, including from predictions, can be decisive to identify possible duplication events.

One could mention that another possible hypothesis would be that gene duplication started from NSm, not from Gn. This is, however, unlikely, since Gn is a structural and thus, necessary, protein; also, there are viruses in *Bunyavirales* for which no NSm was identified, such as in *Hantaviridae*, where Gn^cyto^, however, shows a highly similar double-zinc finger motif as in CCHFV [19,20]. Convergent evolution from a host protein could also be considered as a possible origin; however, as already discussed by Estrada and coworkers [19,20], the Gn, and thus also the NSm, double zinc fingers do not fold independently as classical zinc fingers do, but each finger affects the folding of the other [19,20]. Few intimately contacting double zinc fingers were reported in the PDB, one example being the yeast Bcd1 protein [21], which, however, shows a different 3D fold than CCHFV Gn^cyto^.

For *Phenuiviridae* and *Tospoviridae*, our models suggest that Gn gene fragment duplication was not the origin of NSm, and the protein, with a completely different structural organization than Gn^cyto^, must have been acquired by another, yet undetermined, mechanism.

The Gn^cyto^ zinc fingers are believed to be involved in nucleic acid interactions with the viral genome or the nucleoproteins [10,19,20]. The exact role of NSm in CCHFV infection is still unclear, and whether NSm has evolved for new purposes or rather for supporting certain functions of Gn by increasing their level remains to be determined.

## 5. Conclusions

Based on the current knowledge of the phylogenetic tree of evolutionary relationships among different viral lineages of the order *Bunyavirales* [1], it is unclear how NSm originated and evolved, which makes it difficult to trace its origin. Our analysis, based on structure prediction, suggests that gene fragment duplication events in the *Nairoviridae* and *Peribunyaviridae* families might have led to NSm, while this is clearly not the case for the *Phenuiviridae* and *Tospoviridae* families. Experimental structural data will, however, be needed to confirm both findings. With gene duplication already reported for GP38 and Gc in CCHFV, our work adds NSm as a candidate product from such an event. Duplication of gene fragments has rarely been reported in RNA viruses for proteins whose 3D structures are not available, which may be due to the high mutation rates in these viruses; as a consequence, the resulting low protein sequence similarity can significantly complicate a conservative sequence search approach. Thus, the present analysis not only provides the first structural models for the enigmatic NSm proteins, but also highlights the importance of structure predictions to identify duplications of gene fragments in RNA viruses.

## Figures and Tables

**Figure 1 viruses-16-00090-f001:**
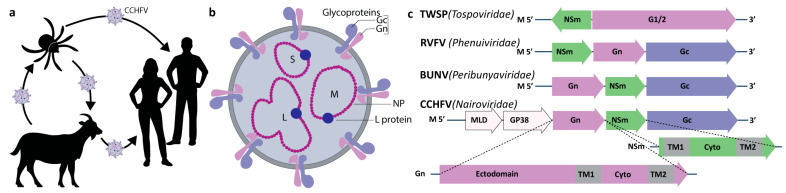
The *Bunyavirales*. (**a**) Transmission modes of *Bunyavirales* using CCHFV as an example, where the virus transits between the tick vector, livestock, and humans. (**b**) The CCHFV viral particle, with the Gn and Gc glycoproteins in pink and purple, respectively. The three L, M, and S genome segments wrapped by the nucleoprotein (NP) are shown in purple, with the L polymerase in blue. (**c**) The NSm non-structural protein is encoded on the M gene of certain *Bunyavirales* species [3].

**Figure 2 viruses-16-00090-f002:**
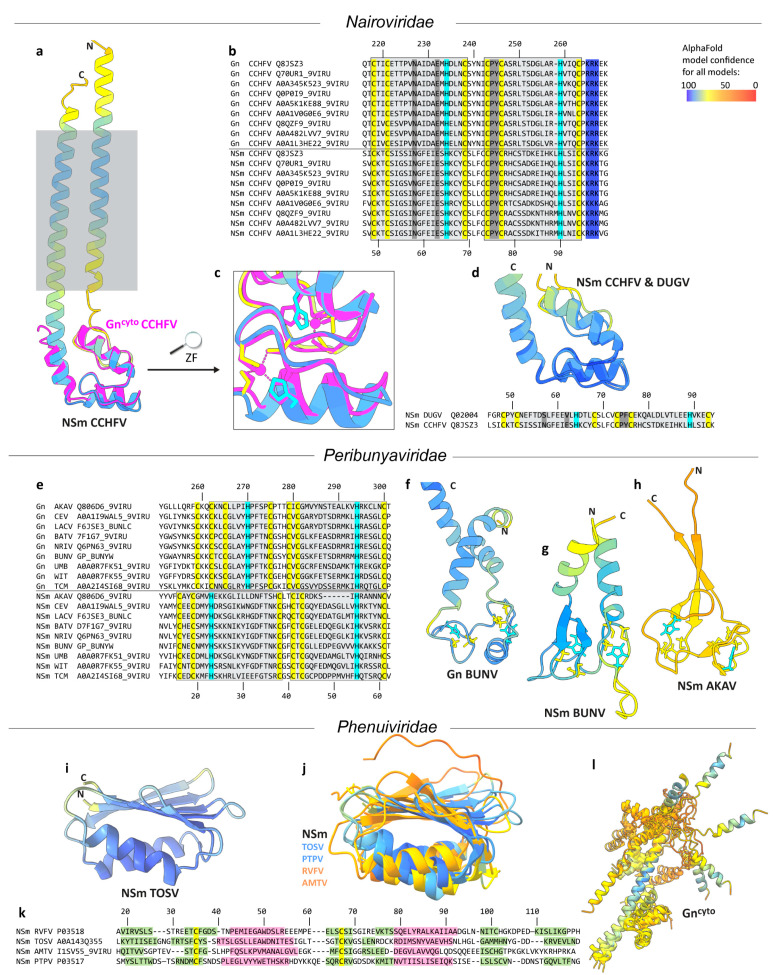
AlphaFold2 structure predictions and sequence alignments of the *Bunyavirales* NSm and Gn^cyto^ proteins. (**a**) CCHFV NSm model overlayed with Gn^cyto^ (PDB 2L7X [10]) model, in magenta, with the grey box indicating the lipid bilayer. Colors reflect the reliability of the prediction as assessed within AlphaFold pLDDT values, from blue (reliable) to red (poor), as shown in the bar at the top right. (**b**) Sequence alignments of Gn^cyto^ and NSm^cyto^ of different CCHFV strains (for the full alignment and numbering of the M polyproteins, see Appendix A). Residue numbering is shown for Q8JSZ3. Conserved residues are highlighted: yellow, Cys; cyan, His; blue, basic residues; grey, other. (**c**) The double ZF motif conserved in NSm and Gn. (**d**) Sequence alignments and AlphaFold2 models of DUGV and CCHFV NSm (for the full CCHFV/DUGV M polyprotein alignment, see Appendix A). (**e**) Gn and NSm sequence alignment for the *Peribunyaviridae* (for the full alignment and numbering of the M polyproteins, see Appendix A). AKAV, Akabane virus; CEV, California Encephalitis Virus; LACV, La Crosse Encephalitis Virus; BATV, Batai Virus; BUNV, Bunyamwera Virus; NRIV, Ngari Virus; UMBV, Umbre Virus; WITV, Witwatersrand Virus; TCMV, Tacaiuma Virus. Structure predictions (**f**,**g**) revealed less similarity between Gn^cyto^ and NSm^cyto^ than in *Nairoviridae*. AKAV NSm (**h**) showed poor prediction accuracy. (**i**) *Phenuiviridae* NSm was predicted in a highly accurate fold for TOSV and PTPV, but less so for RVFV and AMTV (**j**). (**k**) Manual sequence alignment of the NSm domains of different members of the *Phenuiviridae* family taking into account the predicted secondary structures: α-helices, pink; β-strands, green. Conserved Cys are highlighted in yellow. (**l**) Gn^cyto^ model showing poor accuracy and poor similarity to NSm. Colors in all models indicate the AlphaFold2 pLDDT values, from blue (reliable) to red (poor), as shown in the bar at the top.

**Figure 3 viruses-16-00090-f003:**
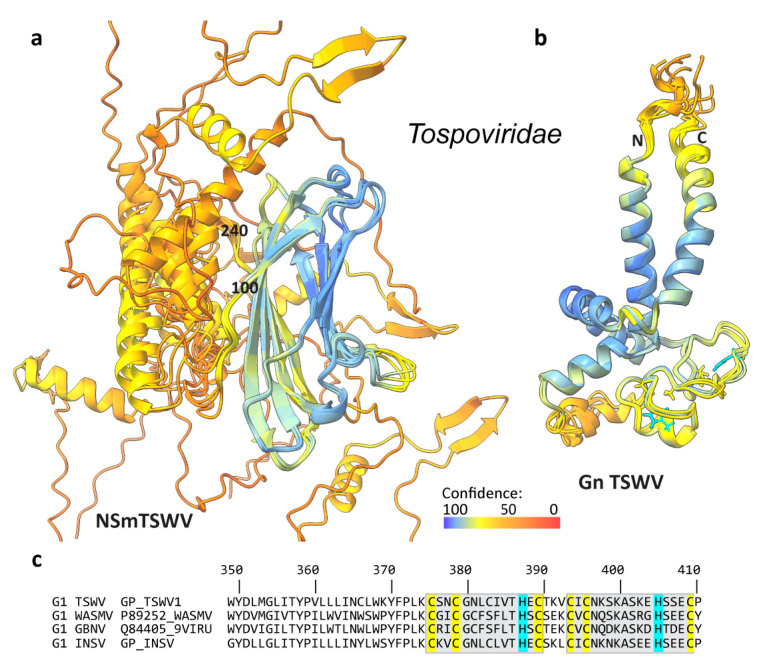
AlphaFold2 structure predictions of the *Tospoviridae* family NSm and Gn^cyto^ proteins, with the example of Tomato Spotted Wilt Virus (TSWV). (**a**) Overlay of the five predicted TSWV NSm models (GenBank accession number S48091, as in [13]). Colors reflect the reliability of the prediction as assessed with the AlphaFold pLDDT values, from blue (reliable) to red (poor), as shown by the bar at the bottom. Numbers indicate the start and end of the β-sheet motif. (**b**) Predicted models for the Gn double ZF motif, with side chains of Cys/His residues binding the zinc ions. An alignment of the sequences, with the motifs indicated, is provided in reference [13]. (**c**) Alignment of the zinc finger motif (for the full alignments, see Appendix A) of four different *Tospoviridae*: TSWV (Uniprot P36291, Genebank S48091), Watermelon Silver Mottle Virus (WSMV, Uniprot P89252, Genebank U75379), Groundnut Bud Necrosis Virus (GBNV, Uniprot Q84405, Genebank U42555), and Impatiens Necrotic Spot Virus (INSV, Uniprot Q01260 and Genebank M74904). Genebank codes are provided to help compare these sequences to those in reference [13].

## Data Availability

All AlphaFold models are available from the corresponding authors on request.

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
