# Peer review of "Do NSm Virulence Factors in the Bunyavirales Viral Order Originate from Gn Gene Duplication?"

_viruses, 2024, doi:10.3390/v16010090_

Round 1

Reviewer 1 Report

Comments and Suggestions for Authors

This manuscript uses AI (here we go!) to analyze the Nsm (non-structural, middle segment) sequences of selected members of the Bunyavirales. The authors found that a globular domain in the derived structure for a tick-borne nairovirus (CCHF)aligns structurally with a domain in the CCHF Glycoprotein N. Similarly, there is some -though less- predicted similarity between the Nsm and Gn proteins of the peribunyavirus BUN. Other bunyavirales proteins are also compared. The authors propose that the Nsm in the cases where there is similarity may have arisen from a gene duplication.

The publication is hypothesis generating and the one experimental figure is clear. The supplemental material is excessive and not digested easily as formatted. I suggest that it be reformatted for clarity, and perhaps deposited in the authors' website and referenced in the paper. 

The abstract associates these findings to the role of Nsm in virulence; this should be modified, as the findings in the paper do not relate to pathogenesis.

Comments on the Quality of English Language

Minor sentence structure issues noted.

Author Response

We would like to thank very much the referee for the very useful and also very mindful comments. We hope that our response is satisfactory.

Reviewer 1 :

This manuscript uses AI (here we go!) to analyze the Nsm (non-structural, middle segment) sequences of selected members of the Bunyavirales. The authors found that a globular domain in the derived structure for a tick-borne nairovirus (CCHF)aligns structurally with a domain in the CCHF Glycoprotein N. Similarly, there is some -though less- predicted similarity between the Nsm and Gn proteins of the peribunyavirus BUN. Other bunyavirales proteins are also compared. The authors propose that the Nsm in the cases where there is similarity may have arisen from a gene duplication.

The publication is hypothesis generating and the one experimental figure is clear. The supplemental material is excessive and not digested easily as formatted. I suggest that it be reformatted for clarity, and perhaps deposited in the authors' website and referenced in the paper. 

>> We prefer to keep it in the SI so that it stays close to the data in Figure 2. I hope the alignments are also useful to the reader. However, we have reformatted Table S1 to make it less prominent and more readable - I agree that this is a data cemetery, but it puts numbers on the colored backbones shown in Figure 2, which might be useful if black/white copies are used. 

The abstract associates these findings to the role of Nsm in virulence; this should be modified, as the findings in the paper do not relate to pathogenesis.

>> Thanks, we removed.

Reviewer 2 Report

Comments and Suggestions for Authors

This manuscript by Lefebvre et al. describes the structure prediction of bunyavirus NSm using AlphaFold and the finding that the Nairoviridae and partly the Peribunyaviridae NSm cytosolic domains structurally resemble that of cognate Gn glycoproteins, despite low amino acid sequence similarity. This finding leads the authors to hypothesise that NSm proteins derived from Gn proteins by independent gene duplication events. When comparing NSm and Gn cytosolic domain structures there was a high degree of conservation and confidence within the Nairoviridae, but only partial or less conservation within other bunyavirus families which led me to doubt the hypothesis of gene duplication. The authors link the NSm cytocolic domain to virulence although there is no evidence for that given that Gn is not necessarily a known bunyavirus virulence factor. As the authors state, there is no direct correlation between the presence of NSm and virulence, i.e. there are bunyaviruses that contain an NSm protein that are not particularly virulent and virulent bunyaviruses that lack NSm. NSm is further not conserved within single bunyavirus families such as the Nairoviridae. Overall, the authors need to add more evidence to their hypothesis and thus, I suggest the following changes to be considered. 

1. Please add a bunyavirus M segment phylogenetic tree and indicate the presence of NSm on each branch to allow the reader to understand how many independent gene duplication events you predict to have taken place. I more detailed discussion around bunyavirus evolution and the role of NSm would add confidence to the manuscript and the gene duplication theory.

2. Please add a structural analysis of Tospovirus NSm cyto vs Gn cyto, as Tospovirus NSm is also a well known virulence factor. 

3. For viruses for which the Gn structure has not been experimentally determined, could you perhaps first model it against existing Gn structures before comparing NSm cyto to Gn cyto? I am not an expert on AlphaFold but there seem to be some limitations to the predictions when the template structure is of lower quality.   

4. I much prefer the manuscript version on bioRXiv, which is easier to follow and gives more relevant background information. I would ask to adapt some of the paragraphs in the current version back to the earlier version.  

Minor comments:

Line 25: Replace "Viral research" by Virology research

Line 31: Change to "Bunyavirales thus represent.."

Line 65: Figure 1 legend: a) change to: "Transmission modes of Bunyavirales using CCHFV as example.."

Line 66: Replace "life stock" by "livestock"; b) add more detail here

Line 79: Unclear what "RMSDs" stands for

Lines 82-84: Indicate that the first line is the paragraph title by adding line spacing or having the title in bold. Add another introduction sentence to the paragraph rather than starting with what Figure 2a shows. You could use this sentence from your bioRXiv manuscript version: "We used AlphaFold to predict the structures of the protein for several members of the Nairoviridae family."

Line 95: Unclear what CCHC motif stands for

Line 195: Replace "nail down" by a more scientific expression

Lines 210/211: This seems like speculation. There is no amino acid sequence similarity between NSm and any arthropod protein. I am not aware of the concept that viral proteins that contribute to virulence in a certain host derive from that host.  

Line 216: I would argue that this is irrelevant

Lines 217/218: RVFV pathology is most closely linked to NSs.

Supplementary tables 2-8: These alignments do not display well in the provided pdf. Please ensure this is addressed in the final manuscript. 

Comments on the Quality of English Language

Overall, the quality of English was good. There were a few sentences that will need to be edited. 

Author Response

We would like to thank very much the referee for the very useful and also very mindful comments. We hope that our response is satisfactory.

Reviewer 2 :

This manuscript by Lefebvre et al. describes the structure prediction of bunyavirus NSm using AlphaFold and the finding that the Nairoviridae and partly the Peribunyaviridae NSm cytosolic domains structurally resemble that of cognate Gn glycoproteins, despite low amino acid sequence similarity. This finding leads the authors to hypothesise that NSm proteins derived from Gn proteins by independent gene duplication events. When comparing NSm and Gn cytosolic domain structures there was a high degree of conservation and confidence within the Nairoviridae, but only partial or less conservation within other bunyavirus families which led me to doubt the hypothesis of gene duplication. The authors link the NSm cytocolic domain to virulence although there is no evidence for that given that Gn is not necessarily a known bunyavirus virulence factor. As the authors state, there is no direct correlation between the presence of NSm and virulence, i.e. there are bunyaviruses that contain an NSm protein that are not particularly virulent and virulent bunyaviruses that lack NSm. NSm is further not conserved within single bunyavirus families such as the Nairoviridae. Overall, the authors need to add more evidence to their hypothesis and thus, I suggest the following changes to be considered. 

  1. Please add a bunyavirus M segment phylogenetic tree and indicate the presence of NSm on each branch to allow the reader to understand how many independent gene duplication events you predict to have taken place. I more detailed discussion around bunyavirus evolution and the role of NSm would add confidence to the manuscript and the gene duplication theory.

>> Such a tree, and info on the presence of NSm, have been published recently in the nice review from Leventhal (Leventhal, Shanna S., Drew Wilson, Heinz Feldmann, et David W. Hawman. « A Look into Bunyavirales Genomes: Functions of Non-Structural (NS) Proteins ». Viruses 13, no 2 (18 février 2021): 314. https://doi.org/10.3390/v13020314.), even if they unfortunately did not detail Tospoviruses. These have been addressed however in another manuscript before in detail: (Silva, M. S., C. R. F. Martins, I. C. Bezerra, T. Nagata, A. C. De Ávila, et R. O. Resende. « Sequence Diversity of NS M Movement Protein of Tospoviruses ». Archives of Virology 146, no 7 (1 juillet 2001): 1267‑81. https://doi.org/10.1007/s007050170090.). We believe that a complete analysis of the presence of NSm in the Bunyavirales would be very interesting. However, we feel this is beyond the scope of this paper.

  1. Please add a structural analysis of Tospovirus NSm cyto vs Gn cyto, as Tospovirus NSm is also a well known virulence factor. 

>> We have added models for NSm and Gn of the TSWV, and discuss them.

  1. For viruses for which the Gn structure has not been experimentally determined, could you perhaps first model it against existing Gn structures before comparing NSm cyto to Gn cyto? I am not an expert on AlphaFold but there seem to be some limitations to the predictions when the template structure is of lower quality.   

>> Since Alphafold learns from the PDB, its Gn models are in a certain way already based on the existing Gn structures. We certainly could construct homology models – but using which proteins? Alphafold finds that HIV-1 NCp7 closest resembles Gn in the Peribunyaviridae actually. Thus, homology models would not present a better hypothesis than the Gn Alphafold models. We feel that this would introduce unjustified bias, and think that structure determination must be the next step to gain better insight.

  1. I much prefer the manuscript version on bioRXiv, which is easier to follow and gives more relevant background information. I would ask to adapt some of the paragraphs in the current version back to the earlier version.  

>> We did revert several parts from the older version to the current version.

Minor comments:

Line 25: Replace "Viral research" by Virology research

>> done

Line 31: Change to "Bunyavirales thus represent.."

>> done

Line 65: Figure 1 legend: a) change to: "Transmission modes of Bunyavirales using CCHFV as example.."

>> done

Line 66: Replace "life stock" by "livestock"; b) add more detail here

>> done

Line 79: Unclear what "RMSDs" stands for

>> done

Lines 82-84: Indicate that the first line is the paragraph title by adding line spacing or having the title in bold. Add another introduction sentence to the paragraph rather than starting with what Figure 2a shows. You could use this sentence from your bioRXiv manuscript version: "We used AlphaFold to predict the structures of the protein for several members of the Nairoviridae family."

>> done

Line 95: Unclear what CCHC motif stands for

>> done

Line 195: Replace "nail down" by a more scientific expression

>> done

Lines 210/211: This seems like speculation. There is no amino acid sequence similarity between NSm and any arthropod protein. I am not aware of the concept that viral proteins that contribute to virulence in a certain host derive from that host.  

>> we removed

Line 216: I would argue that this is irrelevant

>> We removed

Lines 217/218: RVFV pathology is most closely linked to NSs. 

>> We removed

Supplementary tables 2-8: These alignments do not display well in the provided pdf. Please ensure this is addressed in the final manuscript. 

>> We are sorry for that and will make sure that our own PDF version will be used in the SI.

Round 2

Reviewer 2 Report

Comments and Suggestions for Authors

I would like to thank the authors for addressing my comments in detail. I believe the manuscript has much improved. 

Major comments:

- Now that tospoviruses are being discussed in the main text, please add the relevant M segment composition to Fig. 1C and alignments to the supplementary information. Please relabel your new tospovirus figure as Fig. 3 (currently labelled as Fig. 2). 

- From the text it is unclear whether there is a CCHC motif in tospoviruses. Can you please display the relevant region of TSWV NSm in Figure 3 as you have done for other virus families in Figs. 2b/e/k.  

- Discussion: clearly there is a functional need for bunyaviruses that are vector transmitted to encode a NSm protein. For nairoviruses and peribunyaviruses you propose that NSm originated from a gene duplication of Gn. What is your hypothesis for the origin of phenuivirus and tospovirus NSm, especially given that phenuiviruses are closer related to nairoviruses and tospoviruses to peribunyaviruses than nairoviruses and peribunyaviruses to each other?

Minor comments:

- throughout the text the term "CysCysHisCys" motif is used, please then also change "KRK motif" to "LysArgLys" and "CxxCxxxxHxxxxC" to "CysxxCys...". Or alternatively, define CCHC motif as CysCysHisCys when you mention it the first time and then keep as CCHC. 

- In the new paragraph about Tospoviridae please put all the mentioned virus families in italics; please define what is meant by "P/D-L-X, the D-, and the PLA2 motifs" - as per above I would be ok with you following one way of writing this (either as one letter or three letter codes). 

Author Response

We would like to thank the reviewer for her/his patience; we hope that the present results provide a satisfactory first insight also for the tospoviruses. If further questions about this family arise, we will be happy to collaborate to provide further answers, particularly with respect to our core competence, structural biology.

Major comments:

- Now that tospoviruses are being discussed in the main text, please add the relevant M segment composition to Fig. 1C and alignments to the supplementary information. Please relabel your new tospovirus figure as Fig. 3 (currently labelled as Fig. 2). 

>> Thanks for the meaningful remarks, done.

- From the text it is unclear whether there is a CCHC motif in tospoviruses. Can you please display the relevant region of TSWV NSm in Figure 3 as you have done for other virus families in Figs. 2b/e/k.  

>> Done, with full alignments given in Table S10.

- Discussion: clearly there is a functional need for bunyaviruses that are vector transmitted to encode a NSm protein. For nairoviruses and peribunyaviruses you propose that NSm originated from a gene duplication of Gn. What is your hypothesis for the origin of phenuivirus and tospovirus NSm, especially given that phenuiviruses are closer related to nairoviruses and tospoviruses to peribunyaviruses than nairoviruses and peribunyaviruses to each other?

>> We really have no idea at this point, as our knowledge of evolution is also limited. We are currently engaging in a collaboration with an evolutionary biologist, but a more thorough analysis will take some time. While the high similarity between Gn and NSm in Nairo and Peribunyaviruses caught our eye and seems obvious enough to formulate sound hypotheses, we would like to refrain from further hypotheses until more insight is obtained.

Minor comments:

- throughout the text the term "CysCysHisCys" motif is used, please then also change "KRK motif" to "LysArgLys" and "CxxCxxxxHxxxxC" to "CysxxCys...". Or alternatively, define CCHC motif as CysCysHisCys when you mention it the first time and then keep as CCHC. 

>> Done

- In the new paragraph about Tospoviridae please put all the mentioned virus families in italics; please define what is meant by "P/D-L-X, the D-, and the PLA2 motifs" - as per above I would be ok with you following one way of writing this (either as one letter or three letter codes).

>> Done.
